# Universal non-monotonic drainage in large bare viscous bubbles

Casey Bartlett[1,2], Alexandros T. Oratis ®[1,2], Matthieu Santin[1] & James C. Bird ®[1] ✉

Bubbles will rest at the surface of a liquid bath until their spherical cap drains sufficiently to spontaneously rupture. For large film caps, the memory of initial conditions is believed to be erased due to a visco-gravitational flow, whose velocity increases from the top of the bubble to its base. Consequently, the film thickness has been calculated to be relatively uniform as it thins, regardless of whether the drainage is regulated by shear or elongation. Here, we demonstrate that for large bare bubbles, the film thickness is highly nonuniform throughout drainage, spanning orders of magnitude from top to base. We link the film thickness profile to a universal non-monotonic drainage flow that depends on the bubble thinning rate. These results highlight an unexpected coupling between drainage velocity and bubble thickness profiles and provide critical insight needed to understand the retraction and breakup dynamics of these bubbles upon rupture.

From afar, a bubble resting at the surface of a liquid may appear static; however, the liquid film that comprises its cap is typically in motion, draining into the surrounding liquid from a combination of gravitational and capillary forces[1,2]. Bubble drainage is important because it determines how long bubbles remain at the surface, a factor that is relevant to glass manufacturing[3,4], volcanic eruptions[5], and the stability of polymer foams[6]. The drainage dynamics are also important because they influence the fate of the film upon rupture: the film can break up into aerosols that are relevant to health and climate through the transport of pathogens[7–9] and cloud-nucleating particles[10,11]. Alternatively, the film can bend and entrap air that forms smaller bubbles[12] or, if sufficiently viscous, develop a curious wrinkling instability that has received notable attention in part due to its deeper connection to elastic wrinkles[3,13,14]. The ability to predict the dynamics of these processes crucially relies on how the bubble thickness varies spatially at the moment of rupture.

It is often the case that spherical films will adopt a fairly uniform thickness as they drain, a phenomenon that has been exploited to fabricate uniform elastic shells[15]. For small bare bubbles, an initially uniform film will remain uniform as it thins[16]. However, the fate of an initially non-uniform film is less clear, unless the drainage happens to wipe out the memory of initial conditions, leading to universality. Universal drainage profiles have been calculated and observed for large spherical films driven by gravity and resisted by viscosity for a range of boundary conditions at the two film interfaces. If either of the film surfaces is prevented from moving (no-slip), theoretical and experimental studies illustrate that the velocity increases monotonically with the inclination angle $\phi$ and depends on the local thickness $h$[15,17–19]. The resulting thickness profile remains fairly uniform and collectively thins with time as $t^{-1/2}$. However, if the film surfaces are bare (stress-free) and able to move along with the bulk flow, the film is assumed to obtain a velocity that continues to increase monotonically with inclination angle, but no longer depends on the film thickness[3,20]. This independence modifies the universal thickness profile so that it exponentially decays with time, a result that was originally confirmed with interferometry in the pioneering work by Debregeas et al. [3] and has been observed in numerous studies since[14,20–24].

Because the drainage velocity changes from a thickness-dependent to a thickness-independent state, the predicted thickness profiles for a bare bubble changes as well. Still, the predicted thickness profile is found to remain fairly uniform, with the midpoint and base of the bare bubble maintaining a relative thickness of 1.3 and 4 of that at the top[3]. Yet over the decades, there have been observations with bare bubbles that are inconsistent with this shape factor. For example, if the film ruptures at the top and creates a hole, the rate that this hole opens drastically slows down as it moves outward[3,13,14], in contrast to the

[1]Department of Mechanical Engineering, Boston University, Boston, MA 02215, USA. [2]These authors contributed equally: Casey Bartlett, Alexandros T. Oratis. ✉e-mail: jbird@bu.edu

prediction for a uniform film in which the retraction rate is expected to stay the same or accelerate[25–27]. Additionally, when illuminated by white light, a draining bubble of viscous silicone oil exhibits multiple closely spaced Newton rings[3,14], which highlight that the film is noticeably thickening along the arc-length $s$ from the bubble apex (Fig. 1a). Here we investigate whether the accepted universal drainage solution for bare viscous bubbles[3] applies in practice. Ultimately we find that it does not and show that universality is reached through a different, highly non-uniform thickness profile. More fundamentally, we uncover that the rationale for the universal exponential thinning of bare bubbles is predicated on a film velocity that is derived from shear-dominated flow assumptions. We provide an alternative explanation based on an extensional flow that leads to a spatially non-monotonic drainage velocity.

## Results

### Interferometry experiments

To precisely measure how the thickness varies spatially, we illuminate the bubble with a sodium lamp emitting monochromatic light with wavelength $\lambda = 589$ nm and observe the formation of bright and dark fringes (Fig. 1b, Movie S1). Because the film thickness is much smaller than the bubble radius, the bubble film itself acts as a Fabry–Perot interferometer which can be used to calculate the bubble thickness[28,29] (Fig. S1). Specifically, each bright fringe represents a region of constant thickness and the difference in thickness between successive bright fringes can be computed as $\Delta h = \lambda/(2n\cos\beta) = 278$ nm, where $n = 1.4$ is the refractive index of silicone oil and $\beta = 0.714$ is the angle of refraction. The fringes initially propagate relatively quickly from the apex before progressively slowing down. By recording the progression of fringes past the formation of the black film, we can back-track and assign each fringe a thickness. When the ninth-from-last fringe emerges at the bubble apex, the fringes are spaced far enough from one another to accurately track the fringe progression. The thickness at the top of the bubble is $h = 2.50\,\mu$m at this moment in time, which we define as $t = 0$ (Fig. 1c). Minutes later the last bright fringe ($h = 0.28\,\mu$m) appears at the bubble apex and the fringe with thickness $h = 2.50\,\mu$m is located near the midpoint of the bubble, highlighting that the bubble

thickness profile varies by at least an order of magnitude (Fig. 1d). The last nine fringes remain visible on the bubble surface until the bubble spontaneously ruptures, well into the black-film regime (Fig. 1e). This degree of non-uniformity has not previously been reported for draining bubble films near the point of spontaneous rupture and raises the question of whether the quicker drainage associated with elongational flow provides sufficient time for a universal solution to be reached.

### Seeking universality

To determine whether the drainage exhibits universality, we plot the thickness profile $h(s, t)$ obtained from tracking the last nine fringes and test whether there is any memory of initial conditions. Because the film thickness depends on both time and space, we can either plot the thickness as a function of arc-length $s$ for constant contours of time $t$ (Fig. 2a inset) or as a function of time $t$ for constant contours of arc-length $s$ (Fig. 2b inset). If the thickness exhibits universality, it would follow the form $h(s, t) = h_0 H(s) G(t)$, where $h_0 = h(0, 0)$ is the thickness at the apex at time $t = 0$, $H(s)$ is a dimensionless spatial function, and $G(t)$ a dimensionless temporal function. When we normalize the thickness at any location by the thickness at the bubble apex at that time, we observe that the data indeed collapse to a single curve $H(s) = h(s, t)/h(0, t)$ (Fig. 2a). Similarly, when we normalize the

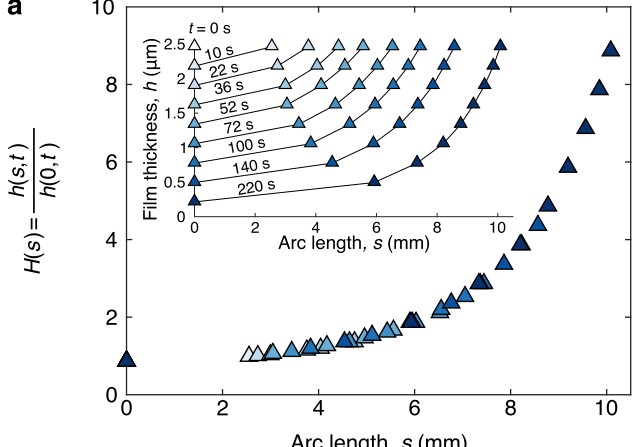

**a**

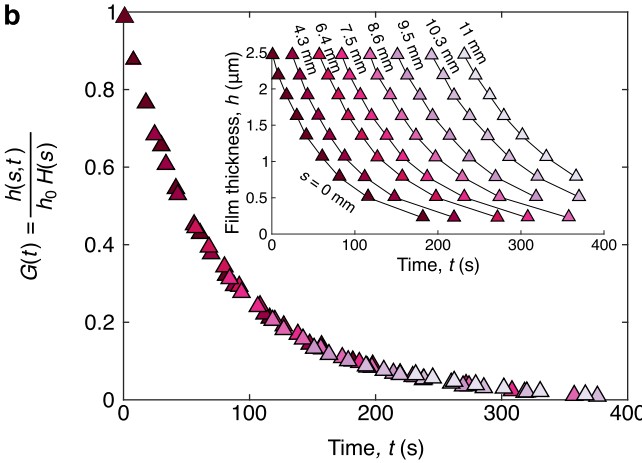

**b**

**Fig. 2 | Spatial and temporal thickness measurements. a** Inner plot: The film thickness increases monotonically with the arc length $s$ with the marker shading denoting different fixed times $t$. Outer plot: These data points collapse onto a single universal curve $H(s)$. **b** Inner plot: The film thickness decreases monotonically with time $t$ with the marker shading denoting different fixed locations $s$. Outer plot: These data points collapse onto a single universal curve $G(t)$. Here $h_0 \equiv h(0, 0) \approx 2.50\,\mu$m. Note that the lines are a guide to the eye and that error bars are excluded, as they would be smaller than the symbol size.

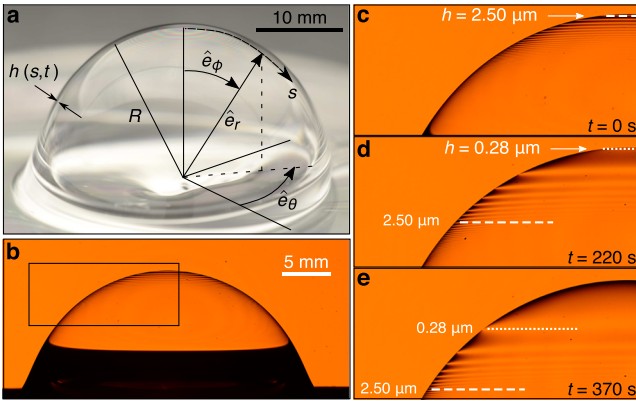

**Fig. 1 | Drainage of a large bubble on the surface of a silicone oil bath. a** When a bubble rises to the bath's surface, it traps with it a thin liquid film of thickness $h(s, t)$, which varies with the arc length $s$ and time $t$. Based on the hemispherical shape of the bubble, the drainage dynamics can be analyzed using a spherical coordinate system in the radial, polar, and azimuthal directions $(\hat{e}_r, \hat{e}_\phi, \hat{e}_\theta)$. **b** Illuminating the bubble with a monochromatic sodium lamp, we observe the appearance of bright and dark fringes. **c** At $t = 0$ s, the ninth to last fringe with a constant thickness of $h = 2.50\,\mu$m appears at the top of the bubble. **d** At $t = 220$ s, the fringe has moved further down the bubble, while the last bright fringe appears at the top with a thickness of $h = 0.28\,\mu$m. **e** At $t = 370$ s, the two highlighted fringes have propagated towards the bubble's base and a black film appears at the apex. Here the bubble radius is $R = 12$ mm and the viscosity is $\mu = 2300$ Pa s.

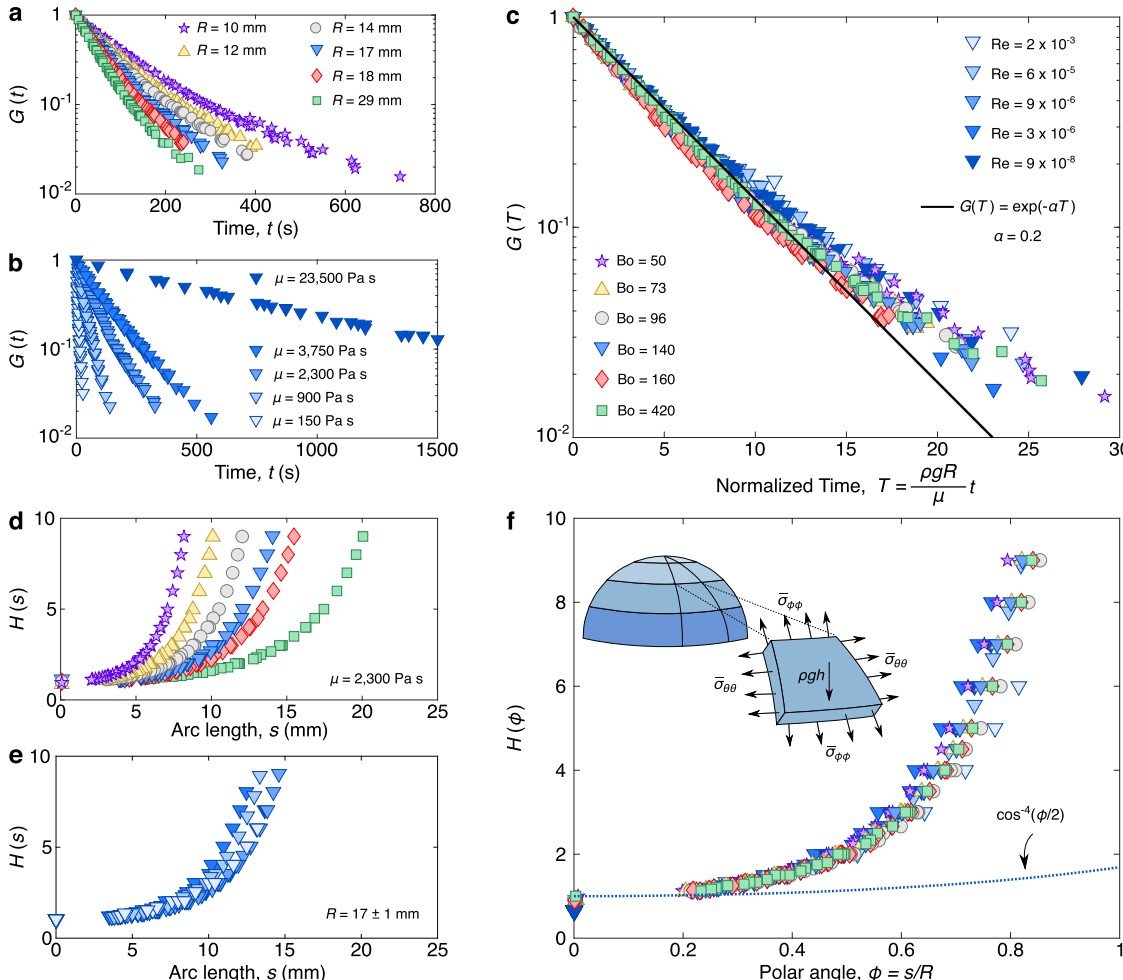

**Fig. 3 | Effects of bubble radius and film viscosity on the temporal and spatial evolution of the film thickness. a** For a constant film viscosity $\mu = 2300$ Pa s, increasing in the bubble radius leads to faster drainage. **b** For a constant bubble radius $R \approx 17$ mm, increasing the film viscosity prolongs the lifetime of the bubble. **c** Plotting the temporal function $G(T)$ against the dimensionless time $T = \rho g R t / \mu$ leads to the collapse of the different data sets onto a single curve. The data follow an exponential decay (solid line) of the form $G(T) = \exp(-\alpha T)$, with a decay rate of $\alpha = 0.2$. **d** The spatial function $H(s)$ varies more gradually along the surface of larger bubbles. **e** Varying the viscosity of the silicone oil does not influence the spatial thickness variation $H(s)$. **f** Switching to the dimensionless polar angle $\phi = s/R$ we observe that the data for the spatial function $H(\phi)$ collapse on top of each other. The existing theory[3] predicts $\cos^{-4}\left(\frac{\phi}{2}\right)$ (dotted line), which underestimates the spatial variation of the thickness. Inset: Schematic illustrating the extensional stresses $\bar{\sigma}_{\phi\phi}$ and $\bar{\sigma}_{\theta\theta}$ in the polar and azimuthal direction, respectively, acting along the bubble walls, which are subjected to the gravitational stress $\rho g h$.

thickness at any time by this empirical shape factor, we again observe that the data collapse to single curve $G(t) = h(s, t)/[h_0 H(s)]$ (Fig. 2b). Therefore, independent of any prescribed kinematics or dynamics, the interferometry results demonstrate the existence of a highly non-uniform universal drainage law.

It has been suggested that surface tension and inertia can critically modify the dynamics of thin viscous films comparable to the bubble films explored here[14,30]. Therefore to evaluate whether surface tension and viscosity influence the drainage of these large, viscous bubbles, we conduct further experiments where we vary the bubble size and the viscosity. Specifically, we consider bubbles with radii in the range 10 mm < $R$ < 30 mm and silicone oil viscosities of $\mu = 150$, 900, 2300, 3750, and 23,500 Pa s. For these liquids, the density and surface tension remains constant with $\rho = 990$ kg m$^{-3}$ and $\gamma = 0.02$ N m$^{-1}$, respectively. Note that these bubble radii are still much larger than the capillary length, such that the Bond number Bo $= \rho g R^2 / \gamma > 50$, where $g$ is the gravitational acceleration. Thus in all cases, the bubble protrudes substantially beyond the surface to form a nearly hemispherical cap[31]. Previous studies on comparable viscous bubbles have observed an exponentially decaying thickness at the bubble apex that was

prolonged by decreasing the radius or by increasing the viscosity[3,14,21–23]. Indeed when $G(t)$ is calculated at all points and plotted on a semi-log scale, these expected trends are observed (Fig. 3a, b). Furthermore, these past studies revealed that for sufficiently large Bond numbers, the apex drainage should follow $h \sim \exp(-\alpha T)$, where $T = \rho g R t / \mu$ is a dimensionless visco-gravitational time and $\alpha$ is a constant dimensionless decay rate. The value of $\alpha$ was theoretically predicted to follow $2\pi R^2/(9S)$, where $S$ is the surface area of the cap[22]. For the large, nearly hemispherical bubbles considered here, this prediction simplifies to $\alpha \approx 1/9$, a value that slightly underestimates other experimental and numerical results acquired at the bubble apex[3,21–23]. Replotting the data (Fig. 3a, b) with the dimensionless time $T$ highlights that our measurements confirm previous observations of a decay rate $\alpha = 0.2$[2] (solid line in Fig. 3c); it also experimentally demonstrates that this exponential drainage extends beyond the bubble apex to other points along the bubble cap.

In contrast to the temporal thinning $G(t)$, which follows the results from previous studies, the shape factor $H(s)$ counters expectation. For all of the bubbles investigated, the shape factor $H(s)$ depends on size $R$, but not on the viscosity $\mu$ (Fig. 3d, e). Nondimensionalizing this data in

terms of the polar angle $\phi = s/R$ allows us to directly compare the experimental results with the theoretical predictions[3], $H(\phi) = \cos^{-4}(\phi/2)$, highlighting the significant discrepancy (Fig. 3f). Furthermore, the collapse of the experimental data reveals that this discrepancy cannot be attributed to surface tension and inertial effects neglected from the model. Specifically dimensional analysis reduces a general function for the universal thickness from $h(s, t, R, g, \mu, \gamma, \rho)$ to $h(\phi, T, \mathrm{Bo}, \mathrm{Re})$, where the Reynolds number $\mathrm{Re} = \rho^2 g R^3/\mu^2$ captures the ratio between inertial to viscous effects subjected to a characteristic visco-gravitational velocity $\rho g R^2/\mu$. Yet, the observed independence of both $G(T)$ and $H(\phi)$ from Bo and Re (Fig. 3c,f) confirms that gravitational and viscous effects alone are responsible for the rate of thinning. Nevertheless, the deviation between the experiments and past theory illustrates that there is a flaw in one of the core assumptions (Fig. 3f). Here we show that the issue arises when tacitly assuming that velocity gradients across the film (shear) dominate velocity gradients along the film (elongation).

## Universal drainage

In a spherical coordinate system, the conservation of mass can be written as

$$\frac{\partial h}{\partial t} + \frac{1}{R \sin \phi} \frac{\partial}{\partial \phi}(hu \sin \phi) = 0, \quad (1)$$

where $u(\phi, t)$ is the drainage velocity in the polar direction $\hat{e}_\phi$ averaged over the film thickness. Motivated by the experimental results of Fig. 3, we seek universal solutions of the form $h(\phi, t) = h_0 H(\phi)G(T)$ and $u(\phi, t) = (\rho g R^2/\mu)U(\phi)V(T)$. The partial differential equation can be separated into a temporal and a spatial ordinary differential equation with a constant parameter $\alpha$:

$$\frac{dG}{dT} = -\alpha V G, \quad (2)$$

$$\frac{H'}{H} = \frac{\alpha}{U} - \frac{U'}{U} - \cot \phi, \quad (3)$$

where the prime denotes differentiation with respect to $\phi$. Provided that $u(\phi, t)$ is known or prescribed, the equations can be integrated to find solutions of $G$ and $H$ subject to the conditions $G(0) = 1$ and $H(0) = 1$ set up by construction. Furthermore, by symmetry, $H'(0) = 0$, which when substituted into Eq. (3) constrains $\alpha$ to be equal to $2U'(0)$. This approach recovers each of the previously published universal thickness profiles $h(\phi, t)$, predicted when the spherical film is constrained by two no-slip boundary conditions[17,18], slip conditions[19,20], two stress-free conditions[3], and mixed conditions along with a time-dependent viscosity[15].

If we assume the viscous dissipation across the film to dominate that along the film, the momentum equation can be solved analytically, yielding a thickness-averaged velocity: $u = -(\rho g/2\mu) \sin \phi[(h^2/12) - (h/2 + b)^2]$, where $b$ is a slip-length[20]. In the no-slip limit ($b = 0$), $u = (\rho g/12\mu)h^2 \sin \phi$, so that—based on our definitions—$U = H^2 \sin \phi$, $V = (h_0^2/12R^2)G^2$, and $\alpha = 2$. Substituting these expressions into Eqs. (2) and (3) reproduce the universal bubble thinning expected for shear drainage in the absence of boundary slip[17,18]. Meanwhile, if the slip is large so that $b = R \gg h$, the velocity field loses its thickness-dependence and reduces to $u = (\rho g/2\mu)R^2 \sin \phi$, so that $U = (1/2) \sin \phi$, $V = 1$, and $\alpha = 1$. These results are identical to those proposed by Debrégeas et al.[3] for the extensional drainage of bare bubbles. However, these velocity prescriptions are valid for shear-based flows, where the immobile surfaces lead to dominant viscous dissipation on the scale of the film thickness. This approach is thus questionable in extensional flows, where the dissipation occurs on the scale of the bubble. Indeed, taking

the limit $b \gg h$ in the shear-dominated equations does not reproduce equations expected for extension-dominated flows[16,32].

## Extensional flow model

To solve for the drainage velocity when film shear is negligible, we note that in the polar direction, conservation of momentum $\nabla \cdot \bar{\boldsymbol{\sigma}} = -\rho \mathbf{g} h$ simplifies to:

$$\frac{\partial \bar{\sigma}_{\phi\phi}}{\partial \phi} + \cot \phi \left( \bar{\sigma}_{\phi\phi} - \bar{\sigma}_{\theta\theta} \right) = -\rho g h R \sin \phi. \quad (4)$$

Here the bulk stress has been integrated over the film thickness to create a 2D surface stress, which we highlight with a bar over the tensor $\bar{\boldsymbol{\sigma}}$ and its diagonal components, $\bar{\sigma}_{\phi\phi}$ and $\bar{\sigma}_{\theta\theta}$. The right-hand side of Eq. (4) represents the weight of the film projected in the polar direction, while the left-hand side represents the viscous resistance arising from the extensional polar stress $\bar{\sigma}_{\phi\phi}$ and azimuthal stress $\bar{\sigma}_{\theta\theta}$ (Fig. 3f inset).

For an incompressible Newtonian fluid subjected to axisymmetric elongation, the viscous stress can be related to the film thickness and velocity with the constitutive relations $\bar{\sigma}_{\phi\phi} = (2\mu h/R)(2\partial u/\partial \phi + u \cot \phi)$ and $\bar{\sigma}_{\theta\theta} = (2\mu h/R)(\partial u/\partial \phi + 2u \cot \phi)$ (see Derivation of governing drainage equations in SI). Note that upon substitution, Eq. (4) becomes the large Bond number limit to the model that Howell developed to investigate the capillary-dominated drainage of small bubbles (see Eq. B38 in ref. [16]). In this limit, each stress has a linear dependence on thickness $h$, as does the film weight. However, the spatial derivative in Eq. (4) prevents $h$ from canceling out of this momentum balance. That is, there is still a contribution of the film thickness in the extensional drainage velocity $u$. As a result, the thickness and the velocity are coupled, in contrast to a shear-dominated momentum balance where the thickness only enters through the boundary conditions.

An exponentially decaying thickness at the bubble apex has been previously attributed to a thickness-independent film velocity arising from plug-flow or near-plug-flow conditions[3,20]. Therefore, it may initially seem surprising that exponential decay could occur in a situation where the drainage velocity depends on the film thickness. Seeking a universal solution to Eq. (4) reveals that the temporal dependence of the thickness profile, $G(T)$, passes unaltered through the spatial derivative and exactly balances itself out of the drainage dynamics. Provided the boundary conditions for the momentum equation are also independent of the film thickness, the drainage velocity varies only spatially, so $V(T) = 1$, resolving the potential paradox. Specifically, from Eq. (2), $G$ reduces to the well-known exponential drainage illustrated in Fig. 3c, even though $U$ remains coupled to $H$.

Rewriting the momentum conservation in terms of the spatial universal solutions $H$ and $U$, we find:

$$\frac{H'}{H}(2U' + U \cot \phi) + (2U' + U \cot \phi)' + \\ + \cot \phi (U' - U \cot \phi) = -\tfrac{1}{2} \sin \phi. \quad (5)$$

Conveniently, the dependence of $H$ can be removed by substituting the right-hand-side of Eq. (3) into the first term of Eq. (5). Therefore the universal drainage velocity, although coupled to $H$, depends only on the velocity boundary conditions and the parameter $\alpha$. By symmetry, $U(0) = 0$; however, for the other boundary at the base of the bubble, the velocity is non-trivial. Howell provides a matching criterion between a lubrication analysis in a viscous bubble cap and a capillary-static solution to the outer boundary[16], yet cautions that this criterion is limited to small bubbles. Specifically, Howell notes that for large bubbles (Bo > 1), the thin-film approximation breaks down in the transition region and questions whether finding an analytic matching solution would even be possible. Thus in the large Bond number limit, it might be tempting to assume that the meniscus height and capillary

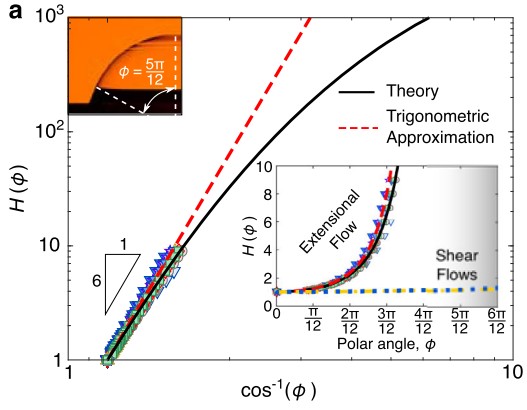

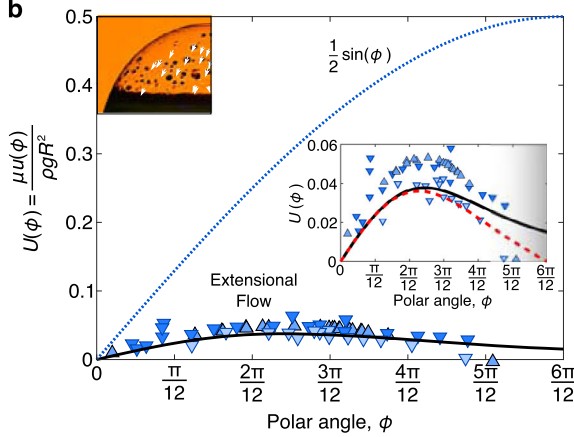

**Fig. 4 | Comparison of experiments and theory for the spatial component of the universal drainage. a** Logarithmic plot of the spatial function $H(\phi)$ against the secant of the polar angle $\phi$. The numerical solution (solid line) nicely matches the experimental data. The spatial function can also be estimated using a trigonometric approximation (dashed line), which follows the scaling $H(\phi) \sim \cos^{-6}\phi$. Inset: In contrast to an extensional flow with two stress-free interfaces (black solid line) that cause a spatial divergence of the thickness, shear flows with one[15] (yellow dashed line) or two no-slip interfaces[17,18] (blue dotted line) lead to a fairly uniform thickness. **b** Plotting the spatial velocity profile $U(\phi)$ against the polar angle $\phi$, we find that the incline plug flow model (dotted line) significantly overestimates the drainage speed. Inset: The numerical model of elongation flow captures the trend of the experimental data for drainage. Note that the marker symbol differentiates the tracer particles ($\Delta$−talcum powder; $\nabla$−glass microbubbles) and marker shading denotes the viscosity following the legend in Fig. 3b.

suction are both negligible so that $U(\pi/2) = 0$; however, the thin-film assumptions used to derive Eq. (5) are inappropriate near this region. To avoid this quagmire, we recognize that we can replace this unknown boundary condition with the previously derived constraint $U'(0) = \alpha/2$. Therefore, Eq. (5) can be solved for a given value $\alpha$, a value we have now directly tied to an unknown velocity constraint at the base of the bubble. Fortunately, the value of $\alpha$ manifests itself in the exponential decay rate of the film thickness; from our experiments (Fig. 3c), along with previous experiments[23] and numerical simulations[22], we conclude $\alpha \approx 0.2$. Indeed, numerically solving for the velocity $U$ using Eq. (5) allows us to directly compute the spatial thickness profile $H$ through Eq. (3). The agreement between the experimental results and the extensional flow model is striking (Fig. 4a). The numerical results indicate that the non-uniformity in thickness continues at higher angles. Even though this prediction cannot be confirmed by our interferometry experiments, our model suggests that large bubbles with two bare surfaces (extensional flow) can be a hundred to a thousand times thicker near the base than at the top. By contrast, the thickness profiles of bubbles with one bare and one rigid surface or two rigid surfaces (shear flow), are still predicted to be nearly uniform and incidentally equivalent (Fig. 4a inset).

The numerical solution to the universal drainage velocity $U(\phi)$ from Eq. (5) is plotted in Fig. 4b. There are two notable features that differentiate this universal bubble drainage profile from the previous theories[3,20], where $U = (1/2)\sin(\phi)$. First, the numerical results predict a maximum velocity that is over an order of magnitude smaller than expected from the natural scaling relation $u \sim \rho g R^2/\mu$. Second, the drainage velocity follows a non-monotonic function with polar angle $\phi$. As a parcel of fluid flows down the bubble it initially accelerates, reaching a maximum velocity at $\phi = 0.61$ radians or 35°. Beyond this angle, the fluid parcel decelerates until it reaches the base of the bubble. Similar velocity profiles and values have recently been observed in numerical studies that investigate the effects of surfactants on surface bubble drainage[24].

To better understand how the coupling of $H$ and $U$ leads to these unexpected results, we analytically approximate the universal solutions to $h(\phi, t)$ and $u(\phi)$ in terms of the parameter $\alpha$. Low-order Taylor or Padé approximations are not particularly effective in approximating the thickness and velocity profiles away from the origin (Fig. S2). Instead, we seek a trigonometric approximation for the thickness

using the form $H(\phi) = \cos^{-m}\phi$, where $m$ is constant. This approximation satisfies the boundary conditions of $H$ and has the added benefit that, when substituted into Eq. (3), an exact corresponding solution to $U$ can be obtained. Furthermore, this solution to $U$ satisfies the boundary condition $U(0) = 0$ and the approximation $U(\pi/2) = 0$. Matching the curvature of $H$ at the bubble apex, we find that $m = 1 + 1/\alpha$, leading to the approximations:

$$h(\phi, t) \approx h_0 \left[ \cos^{-(1+\frac{1}{\alpha})}\phi \right] \left[ \exp\left( -\alpha \frac{\rho g R t}{\mu} \right) \right], \quad (6)$$

$$u(\phi) \approx \frac{\rho g R^2}{\mu} \left[ \alpha^2 \left( \frac{\cos\phi - \cos^{(1+1/\alpha)}\phi}{\sin\phi} \right) \right], \quad (7)$$

with $\alpha \approx 0.2$ when $Bo \gg 1$. To be clear, $\alpha$ is not a fitting parameter. It is set by the boundary conditions, and we are confident of its value. We have chosen to keep the equations general to illustrate how the boundary condition impacts both the universal thickness and drainage velocity profiles. The trigonometric approximations quantitatively capture the unexpected features observed in the numerical results (Fig. 4). The degree of non-uniformity of the film thickness near the bubble apex is linked to $\alpha$ so that with $\alpha = 0.2$, the thickness near the apex increases as $1/\cos^6(\phi)$, a finding that agrees well with the numerical and experimental results. Additionally, both the peak velocity and the angle where this maximum occurs depend on $\alpha$. Indeed, the prediction that this velocity peaks at just 4% of its characteristic scaling can be directly attributed to $\alpha$ having a value of 0.2.

To evaluate our theoretical predictions for the drainage velocity, we conduct further experiments with microscopic solid particles embedded into the silicone oil (Fig. 4b inset, Movie S2). By tracking the positions of the particles as they flow down the bubble, the Eulerian drainage velocity can be reconstructed (Fig. 4b). As predicted, the drainage speeds collected at different times and for different viscosity films all collapse onto a single dimensionless curve that is a non-monotonic function of polar angle $\phi$. Similar non-monotonic velocity profiles were obtained with talcum powder and 20 μm diameter glass microbubbles as tracers, suggesting that any influence of the particles on the velocity field was small. Despite the uncertainty in the speed measurements, both the numerical solution and approximation follow

a similar trend with the experimental data, in stark contrast to the previously assumed profile (Fig. 4b).

In addition to supporting the model, the velocity measurements provide direct evidence that the film thickness continues to increase at polar angles well beyond the limits of our interferometry data. Velocity measurements extend to polar angles of $\phi \approx 5\pi/12$, at which point the non-uniformity of the film is sufficient to scatter the transmitted light (Fig. 4 top-left inset). Up until that angle, the numerical and trigonometric approximation curves for $U$ (red dashed and black solid curves in Fig. 4b) are consistent with the measured velocity data. Because conservation of mass links these curves to the corresponding ones for $H$ (red dashed and black solid curves in Fig. 4a), we conclude that the theoretical models approximate the thickness up to at least $\phi \approx 5\pi/12$.

## Discussion

We anticipate our results will be relevant to a range of phenomena in which large bubbles undergo extensional or near extensional flow. The degassing of bubbles in liquids, including molten glass and lava, is controlled by the bubble drainage dynamics[3], and the drainage rate has been used to probe the mobility of various liquid interfaces[19,20,24]. In addition, the dynamics upon rupture, such as whether a film aerosolizes[33], wrinkles[14], or traps smaller bubbles[12], depend on the local thickness. Our results demonstrate that the drainage and thickness profiles dramatically differ from expectations, and unlike shear flow, the shape of these profiles is coupled to the temporal dynamics through the parameter $\alpha$. Although our analysis focuses on large, bare bubbles, we also anticipate that these results would extend to moderate-sized bubbles or those with mobile surfactants for which exponential drainage has previously been observed[20,22,24]. For these bubbles, the value of $\alpha$ increases with decreasing the Bond number. Furthermore, our results counter the idea that $\alpha$ depends on a slip boundary condition for near-plug flow. Instead, we directly link $\alpha$ to the rate that capillarity near the bubble base can adsorb the falling liquid and conclude that this rate must be independent of the film thickness for exponential drainage to be observed.

The existence of universal non-monotonic drainage can also be exploited to probe and fabricate highly non-uniform thin shells. Because the thickness profile is non-uniform, interference fringes remain visible on the side of the bubble long after a black film develops, and due to universality, the positions of these fringes reveal how quickly the top of the bubble is draining along with its thickness at spontaneous rupture. At the late stages of the bubble lifetime, the fringe locations indicate that the top of the bubble is ~100 nm thick (Fig. 1e) and the exponential drainage rate decreases (Fig. 3c). This deviation can be attributed to strain hardening due to polymer stretching[34] or disjoining pressure[35], effects which are no longer subdominant as the bubble thickness reaches submicron scales. Our analysis can be extended to consider more complex extensional rheology, including the effects of surfactant concentration on the thinning rate $\alpha$[24], or a time-varying viscosity through nontrivial solutions to $V(T)$. A time-varying viscosity has been used to fabricate elastic shells with nearly uniform profiles when spherical film flowed under shear while simultaneously curing[15]. Our results provide an analogous pathway to fabricate highly non-uniform elastic shells by combining curing and the extensional flow of a large draining bubble.

## Methods

Bubbles were created by injecting air into various silicone oil baths. The supplementary information contains further detail on the methods used to experimentally measure the bubble film thickness, the bubble drainage velocity, and the silicone oil viscosity. A significant difference was observed between the measured viscosity and that reported on the manufacturer's label (Table S1), emphasizing the need for independent viscosity measurements.

## Data availability

The experimental raw data supporting the findings of this study are available from the corresponding author upon request. The processed data are available within the manuscript and its Supplementary Information Files. Source data are provided with this paper.

## Code availability

The code used to numerically solve for the velocity and thickness is available from the corresponding author upon request.

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

## Acknowledgements

The authors acknowledge partial funding from the National Science Foundation under Grant No. 1351466 (J.C.B.) and from the 2012 ENS Paris-Saclay internship (M.S.). We appreciate the assistance of Guanshi Li and Timothy Farmer for their contributions to this project in 2012–2013.

## Author contributions

J.C.B. conceived the project. C.B., A.T.O., and M.S. designed the experiments and analyzed the data. C.B., A.T.O., and J.C.B. developed the theoretical model. C.B., A.T.O., and J.C.B. wrote the manuscript.

## Competing interests

The authors declare no competing interests.
