## [Peer Review File · Nature Communications]

Universal non-monotonic drainage in large bare viscous bubblesREVIEWER COMMENTS

Reviewer #1 (Remarks to the Author):

The authors measure the thickness and velocity profiles in a large viscous bubble and establish the equations governing the flow induced by the gravity. They decouple the temporal and spatial evolutions of the film thickness and velocity, and were able to establish an equation involving only the spatial component of the velocity noted U . The problem is closed with boundary conditions and an adjustable parameter. The equation is then solved using various analytical expansions and using numerical simulations. The spatial and temporal evolution of the velocity and thickness fields are deduced from U and compared to the experiments, with a very good agreement. One saillant point is the non-monotonic variation of the velocity from the top to the bottom of the bubble which is evidenced here for the first time.

The paper cites different papers adressing very different situations: drainage of a thin film on a spherical solid, drainage of bubble with surfactants or traces of surfactants, and drainage of a bubble of pure liquid. The work discussed in the present paper is only relevant to the third case.

To my best knowledge, the measure of the thickness profile is new for this system, but is not technically new. This is the same for the velocity profile.

The physical ingredients controlling the flow in the situation of interest are already identified in a previous paper by Debregeas et al (citation 2) (Note that there is a clear flaw in this paper by Debregeas et al., as discussed in Oratis et al (citation 20), but related to the dynamics after the bubble rupture, not to the drainage process discussed here).

The analytical resolution is certainly improved compared to citation 2 in wich only a partial and qualitative scaling was proposed.

The obtained agreement between the theoretical and experimental values of the thickness and velocity as a function of time and space, for many viscosities and many bubble sizes is excellent.

The paper is clearly written, even if the separation between the SI and the main paper disturbs the reading of the theoretical part, and the references to the previous models are rather evasive.

The results are thus clearly new and interesting, and represent an impressive and reliable work but maybe slightly to incremental to deserve publication in Nature Com.

Remarks

1) typo : lenthscale

2) Fig. 1 : the hat notation is not clear (variable or unit vector)

3) change "an exponentially decaying drainage at the bubble apex " for "an exponentially decaying thickness at the bubble apex"

4) "the shape factor $H(s)$ defies expectation" is maybe exaggerated

5) "Over the range bubbles investigated" is unclear

6) Typo in "the drainage dynamics of viscous bubbles. (Fig. 3f). "

7) "without prescribing the standard monotonic incline flow." should be much more explicit. What are the assumptions of this model, what are the solutions ?

More generally two reference cases are used, for which a clear definition should be given, as well as the relevant properties: this standard monotonic incline flow and the shear flow. In each case, it should be clearly said if the model is valid or not and for which physical assumptions (see also the related points 10, 11, 12).

8) " Importantly, all of these previous examples prescribed a drainage velocity that increased monotonically with polar angle as $u \approx \sin(\phi)$. " This is not entirely clear to me. Especially, with a Poiseuille flow (cases no slip and slip) the velocity depends on the thickness, which may introduce an additional dependency on ϕ .

9) " Note that upon substitution, (4) becomes the large Bond number limit to the model that Howell [3] developed to investigate the capillary-dominated drainage of small bubbles" . The reference seems to be 2D : please check the reference or explain more precisely the relationship.

10) "However, the spatial derivative in (4) prevents h from canceling out of this momentum balance, as was previously assumed" . Please provide a precise reference associated to "as was previously assumed".

11) Fig 4. How is obtained the shear flow solution in the inset ? Provide a reference or an explanation.

12) "Remarkably, replacing the shear flow with an elongation flow causes the thickness profile to rapidly diverge, leading to a highly non-uniform universal thickness profile." I do not understand why the authors underline this point, as I expected indeed a very different results for both flows. The main message concerning the thickness field seems to me to be the result of fig. 3f, where the results of ref 2 is shown to be wrong.

13) " as demonstrated by Howell [5]" (in SI) : please refer to a precise equation.

Reviewer #2 (Remarks to the Author):

The paper untitled “Universal non-monotonic drainage in large viscous bubbles” by C. Bartlett et al. presents experiments and modelling on the liquid drainage in the film of a surface bubble.

The experimental data of film thinning are very nice and cover a large set of parameters (bubble size, viscosity and position in the film). Thanks to all these experimental results the authors show that the film thickness is not uniform but it increases from the top to the bottom over more than an order of magnitude, and that the thinning is not monotonic. They identify a maximal velocity for a polar angle around $\pi/4$. Importantly, these results show that assumptions of previous model are wrong, thus, the authors propose a new model, which captures very well the experimental results. For all these reasons I would recommend publication of the paper. Nevertheless, there some points that I would like to be corrected, explained, adjusted ...

1- In the introduction, the literature review seems misleading, a large part of the introduction is concerning bubbles that are not in the same regime of the present study (bubbles with surfactant, small bubble, anti-bubble, low viscosity ...) and it is not sure that a non specialist would make the difference. I would recommend preventing any amalgam and to highlight that there is a large range of parameters in the case of bubble surface that would lead to different physics. The numerical work of O. Atasi (Langmuir 2020) who showed non monotonic velocity evolution with polar angle (at low surfactant concentration) is missing.

2- The authors write that their results “are consistent with these previous observations with a decay rate $\alpha = 0.2$ ”, but these value of decay rate was previously reported in the review Miguet et al. thus instead of consistent, I would say that it confirms previous results.

3- Finally, the interferometry method is used successfully in a range of film from 0.28 to 2.5 μm so only one decade, but the authors show normalized graphs that vary over two decades, it would be interesting to comment this point.

4- In table one the lower viscosity is equal to 180 Pa.s whereas it is 150 in the text.

Responses to Reviewers:

We thank the Editor and Reviewers for their insightful feedback, which we have used to update our manuscript accordingly.

Below are point-by-point responses for each of the reviews.

Reviewer 1:

The authors measure the thickness and velocity profiles in a large viscous bubble and establish the equations governing the flow induced by the gravity. They decouple the temporal and spatial evolutions of the film thickness and velocity, and were able to establish an equation involving only the spatial component of the velocity noted U . The problem is closed with boundary conditions and an adjustable parameter. The equation is then solved using various analytical expansions and using numerical simulations. The spatial and temporal evolution of the velocity and thickness fields are deduced from U and compared to the experiments, with a very good agreement. One saillant point is the non-monotonic variation of the velocity from the top to the bottom of the bubble which is evidenced here for the first time. The paper cites different papers adressing very different situations: drainage of a thin film on a spherical solid, drainage of bubble with surfactants or traces of surfactants, and drainage of a bubble of pure liquid. The work discussed in the present paper is only relevant to the third case. To my best knowledge, the measure of the thickness profile is new for this system, but is not technically new. This is the same for the velocity profile. The physical ingredients controlling the flow in the situation of interest are already identified in a previous paper by Debregeas et al (citation 2) (Note that there is a clear flaw in this paper by Debregeas et al., as discussed in Oratis et al (citation 20), but related to the dynamics after the bubble rupture, not to the drainage process discussed here). The analytical resolution is certainly improved compared to citation 2 in wich only a partial and qualitative scaling was proposed. The obtained agreement between the theoretical and experimental values of the thickness and velocity as a function of time and space, for many viscosities and many bubble sizes is excellent.

The paper is clearly written, even if the separation between the SI and the main paper disturbs the reading of the theoretical part, and the references to the previous models are rather evasive. The results are thus clearly new and interesting, and represent an impressive and reliable work but maybe slightly to incremental to desserve publication in Nature Com.

We thank the reviewer for their thorough comments. In addressing these comments, in particular the connection to previous drainage models, we believe that our paper has significantly improved. Below are our responses to each remark raised by the reviewer.

1. *typo : lenthscale*

We thank the reviewer for pointing out this mistake, and we have changed it to “*lengthscale*”.

2. *Fig. 1 : the hat notation is not clear (variable or unit vector)*

We agree with the reviewer that the definition of the unit vectors was not clear. We have altered the figure, which now includes the three unit vectors represented as e_r , e_ϕ , e_θ .

3. *change "an exponentially decaying drainage at the bubble apex " for "an exponentially decaying thickness at the bubble apex"*

We have changed the specific sentence to “*an exponential decaying thickness at the bubble apex*”.

4. *"the shape factor $H(s)$ defies expectation" is maybe exagerated*

We agree with the reviewer that the verb “*defies*” is perhaps a bit excessive. We have changed the sentence to “*the shape factor $H(s)$ counters expectation*”.

5. *"Over the range bubbles investigated" is unclear*

Indeed we were missing a word. The sentence now reads “*For all of the bubbles investigated*”.

6. *Typo in "the drainage dynamics of viscous bubbles. (Fig. 3f). "*

We thank the reviewer for pointing out the typographical error. We have removed the unintentional period.

7. *"without prescribing the standard monotonic incline flow." should be much more explicit. What are the assumptions of this model, what are the solutions? More generally two reference cases are used, for which a clear definition should be given, as well as the relevant properties: this standard monotonic incline flow and the shear flow. In each case, it should be clearly said if the model is valid or not and for which physical assumptions (see also the related points 10, 11, 12).*

We agree with the referee and have modified the text and figures to be more explicit about the various flows that we reference. We have removed all references to a standard incline flow. Instead in the paragraph following equations (2) and (3), we provide explicit definitions of the velocity profiles and boundary conditions for the two main referenced cases (gravity-driven

viscous bubble drainage) with two no-slip boundary conditions and two stress-free boundary conditions.

In Figs. 3f and 4b, we removed the label “inclined flow” and instead have focused on the functional form of the profiles. In Fig. 4a, we plot both the universal thickness solutions for two boundary conditions that lead to shear flow (two no-slip interfaces; one no-slip and one stress-free interface), noting that they are actually equivalent.

8. *" Importantly, all of these previous examples prescribed a drainage velocity that increased monotonically with polar angle as $u \approx \sin(\phi)$. " This is not entirely clear to me. Especially, with a Poiseuille flow (cases no slip and slip) the velocity depends on the thickness, which may introduce an additional dependency on ϕ .*

We thank the reviewer for this comment. Indeed, for no-slip boundary conditions at the interface, the velocity depends on the thickness and, as a result, introduces an extra dependency on ϕ . Because the viscous dissipation for no-slip interfaces occurs on a lengthscale of the thickness, the equations for u and h are decoupled. As a result, one can get the velocity profile from the momentum equation and then solve for the thickness using the conservation of mass equation. For two no-slip boundary conditions (Couder et al. 2005, Lhuissier and Villermaux 2012) and a no-slip/stress free boundary condition (Lee et al. 2016), the spatial dependence of the thickness from the apex to the base is $H = 1$ ($\phi = 0$) to $H \sim 1.3$ ($\phi = \pi/2$). Therefore, the dependence of the drainage speed on the polar angle is primarily due to the $\sin(\phi)$ term. Nevertheless, to avoid any confusion, we have removed the last part of the sentence (“*as $u \approx \sin(\phi)$* ”).

9. *" Note that upon substitution, (4) becomes the large Bond number limit to the model that Howell [3] developed to investigate the capillary-dominated drainage of small bubbles". The reference seems to be 2D : please check the reference or explain more precisely the relationship.*

Indeed, the main focus of the paper by P. Howell (now ref [16]) focuses on a 2D geometry. However, in the Appendix B3, Howell provides the axisymmetric equations for a spherical geometry (Eq. B38), which are the model equations we are referencing here. Note that there appears to be some typographical errors in Eq. B38 (*cos* instead of *cot* and a negative sign missing). Nevertheless, we have now provided the reference to this equation derived by Howell in our manuscript.

10. "However, the spatial derivative in (4) prevents h from canceling out of this momentum balance, as was previously assumed". Please provide a precise reference associated to "as was previously assumed".

We have removed the wording "as was previously assumed". We were initially referring to the work by Debrégeas (as well as others who build on their ideas) who used the thickness independence to justify an extensional flow that leads to an exponential decay. We have clarified these ideas in our revised version by pointing out the difference between a shear-based approach versus an extensional approach.

11. Fig 4. How is obtained the shear flow solution in the inset ? Provide a reference or an explanation.

We agree with the reviewer that more precise details on the shear flow curve in the inset are needed. We have added the following sentence in the caption of Figure 4:

"In contrast to an extensional flow with two stress-free interfaces that cause a spatial divergence of the thickness, shear flows with one [15] or two no-slip interfaces [17,18] lead to a fairly uniform thickness."

In Figure 4, we have added both types of shear flows from the studies of Couder et al. (2005) (Ref [17]) and Lee et al. (2016) (Ref [15]). In particular:

For two no-slip boundary conditions Couder et al., Phys. Rev. Lett. (2005) and Lhuissier and Villermaux, J. Fluid. Mech. (2012) report that:

$$H(\phi) = \left(\frac{4}{3 \sin^{4/3} \phi} \int_0^\phi \sin^{1/3} x dx \right)^{1/2}$$

For no-slip/stress-free boundary conditions Lee et al., Nat. Commun. (2016) show that:

$$H(\phi) \approx 1 + \frac{\phi^2}{10}$$

Although we are not aware of it being explicitly noted in the literature, the solutions for H should actually be the same. Specifically, the difference in the average velocity can be absorbed into the temporal part of the universal velocity V , leading to an identical differential equation for the shape factor. Indeed, the first term in a Taylor expansion for the Couder solution is equivalent to the approximation to the Lee solution. We have noted this equivalence in our text.

12. *"Remarkably, replacing the shear flow with an elongation flow causes the thickness profile to rapidly diverge, leading to a highly non-uniform universal thickness profile." I do not understand why the authors underline this point, as I expected indeed a very different results for both flows. The main message concerning the thickness field seems to me to be the result of fig. 3f, where the results of ref 2 is shown to be wrong.*

We thank the reviewer for the comment, as it highlights that we had not made a strong enough connection between shear and extensional results. We have removed this particular sentence, yet have added significant text to highlight why we believe the comparison between shear and extensional flow to be important. We certainly agree that a central point of our paper is that the thickness field calculated by Debregeas (now ref [3]) is wrong. However, what we find remarkable is that the same tacit assumptions – plug flow built on shear flow with considerable slip – led not only ref [3] astray, but subsequent analysis as well (eg ref. [19]). Indeed, another central message in our paper (and one that we believe gives it a broader appeal), is that the well-known exponential thinning at a bubble apex for plug and near-plug flow has been incorrectly explained. In particular, this explanation is based on universal film drainage profiles in which the velocity continues to increase and is fully independent of the instantaneous film thickness. It has been argued the transition between the shear and extensional flow can be continuously varied by just changing the curvature of the parabolic flow in the draining bubble (or in other words the magnitude of the slip length) without modifying the momentum equation. The consequence is that the universal thickness profile changes slightly, but is still nearly uniform in both shear and extensional flows driven by gravity and resisted by viscosity. Therefore in this context, it is not surprising that there is a very different thinning rate for the two flows, but it is quite remarkable and unexpected that replacing the shear flow with an elongation flow would change the drainage from nearly uniform to one that rapidly diverged. Our paper identifies the core misconceptions and explains how the plug flow will lead to the observed universal exponentially thinning, even when the velocity is coupled with the film thickness and ultimately non-monotonic.

13. *" as demonstrated by Howell [5]" (in SI) : please refer to a precise equation.*

We thank the reviewer for inquiring about the exact equations from P. Howell's work [5]. We have added the text "*As demonstrated by Howell (see Eqs. (75)-(76) in [5])*" in the supplementary information.

Reviewer 2:

The paper untitled “Universal non-monotonic drainage in large viscous bubbles” by C. Bartlett et al. presents experiments and modelling on the liquid drainage in the film of a surface bubble. The experimental data of film thinning are very nice and cover a large set of parameters (bubble size, viscosity and position in the film). Thanks to all these experimental results the authors show that the film thickness is not uniform but it increases from the top to the bottom over more than an order of magnitude, and that the thinning is not monotonic. They identify a maximal velocity for a polar angle around $\pi/4$. Importantly, these results show that assumptions of previous model are wrong, thus, the authors propose a new model, which captures very well the experimental results.

We thank the reviewer for their positive feedback. Below we address each of the points the reviewer raised.

- 1. In the introduction, the literature review seems misleading, a large part of the introduction is concerning bubbles that are not in the same regime of the present study (bubbles with surfactant, small bubble, anti-bubble, low viscosity ...) and it is not sure that a non specialist would make the difference. I would recommend preventing any amalgam and to highlight that there is a large range of parameters in the case of bubble surface that would lead to different physics. The numerical work of O. Atasi (Langmuir 2020) who showed non monotonic velocity evolution with polar angle (at low surfactant concentration) is missing*

We appreciate the reviewer pointing us to the numerical work by O. Atasi et al. and now reference it. In light of the reviewer’s feedback, we have restructured our introduction to focus on the role of boundary conditions on the universal drainage of large viscous bubbles. We hope that our revision more clearly links previous work on shear and elongational drainage, regardless of the exact details of the system of interest.

- 2. The authors write that their results “are consistent with these previous observations with a decay rate $\alpha = 0.2$ ”, but these value of decay rate was previously reported in the review Miguet et al. thus instead of consistent, I would say that it confirms previous results.*

We thank the reviewer for suggesting this particular rewording. We have now changed the “is consistent with” with “confirms”.

3. *Finally, the interferometry method is used successfully in a range of film from 0.28 to 2.5 μm so only one decade, but the authors show normalized graphs that vary over two decades, it would be interesting to comment this point.*

We appreciate the reviewer's keen eye and agree that others might wonder where this 'extra decade' for $G(T)$ comes from. It is a consequence of tracking the interference fringes along the surface of the bubble, rather than just at the top. We have added the following comment on this point in the supplementary information:

Note that in our study, we have limited our analysis to the last nine fringes. The approach we have taken could certainly be extended to include the spatial and temporal resolution to confidently discriminate far more fringes. Additionally, we have chosen to define $H(s)$ based on fringes that we can directly observe. This definition limits the span of H to the number of fringes tracked. By contrast, we define $G(t)$ so that it depends on h and $H(s)$. By including our data for $H(s)$ --- as opposed to only normalizing h by $h(s,0)$ --- we extend the range of $G(t)$ by an extra order of magnitude. An alternative analogous definition for $H(s) = h(s,t)/[h_0 G(t)]$ could be constructed, which would have the benefit of extending its range. We have chosen to avoid this definition here, for one that is more traditional and direct.

4. *In table one the lower viscosity is equal to 180 Pa.s whereas it is 150 in the text.*

We thank the reviewer for pointing out this inconsistency. Indeed the value should have been 150 and have now changed it in the table located in the supplementary information.

REVIEWER COMMENTS

Reviewer #1 (Remarks to the Author):

In the revised version, the connection to previous drainage models is improved and is now more explicit, but still needs clarifications.

The important contribution of the paper is to show that one former model, corresponding to the bare interfaces, was wrong in the literature, and to build the right one. It also provides comparison with other, valid, drainage models, with some interfacial stress at at least one interface.

In the present form, it is difficult for the reader to know which model of the literature is confirmed (the one with interfacial stress) and which one is proved wrong (the one without interfacial stress).

I will recommend the paper for publication if the following remarks are taken into account.

* The introduction would be clearer if the wrong model of the literature was not presented as a fact.

For example in

"In these cases, the thickness-averaged drainage velocity u increases monotonically with inclination angle ϕ and depends on the boundary conditions at the two film surfaces."

Similarly in "Still, the thickness profile remains fairly uniform, with the midpoint and base of the bare bubble maintaining a relative thickness of 1.3 and 4 of that at the top [3]."

is it true ? false ? experimental ? theoretical ? for which case ?

My suggestion is that, at the various places where it is needed, the case with shear was discussed first, as a statement, and then the bare interface case with the experimental results (e. g. exponential decay at the apex) presented as robust, and the model (quasi uniform thickness, monotony of the velocity) presented using expressions as 'has been predicted/ found / assumed to be ...' or equivalent

* In "the film velocity will depend on the local thickness h and adopt a nearly uniform thickness profile" a word is missing

* In "Importantly, all of these previous examples prescribed a drainage velocity that increased monotonically with polar angle."

here again do not mix what is valid and what is not, what is measured and what is assumed.

* "Because this drainage velocity is identical to the extensional flow proposed by Degregeas et al. [3], the universal bubble thinning expected for extensional drainage is recovered."

This sentence is very unclear to me.

For me the important messages are clearly identified in your rebuttal letter (answer 12) but not explicit here, especially the second point below.

In my opinion the discussion should be that :

- the limit " $b=R$ " leads to the Degregeas equations, with a reference to the sentence of the introduction

"Here we show that the issue arises when tacitly assuming that velocity gradients across the film (shear) dominate velocity gradients along the film (elongation)." which is otherwise a little bit mysterious.
- unfortunately taking a limit of large b (or, much better, a small interface elasticity or viscosity) in the shear dominated equation, does not lead to the (proper) extension dominated equation (with a reference, that could be "The drainage of a foam lamella", Breward and Howell). So a very large b is not a physical limit.

As a personal note, I think the notion of slip length is nonsense for liquid-gas interfaces, simply because it should control a velocity difference between the two phases, and the gas phase velocity does not play any role in the dynamics, and is not even discussed. So I strongly support your fight against it.

* in " the film thickness continues to diverge at polar angles ..." I would say to increase

* in "a result that is consistent with an increase in extensional viscosity due to strain hardening."
If a strain hardening is expected or has been measured add a reference. Disjoining pressure is also a reasonable candidate.

* Fig 4a Inset: "In contrast to an extensional flow with two stress-free interfaces that cause a spatial divergence of the thickness, shear flows with one [15] or two no-slip interfaces [17, 18] lead to a fairly uniform thickness."
please refer to the color used for each.

* Fig 3 f : the mention of the "incline flow" said in the rebuttal letter to have been removed is still there.

Supp Mat

* " intensity is measured at an incident angle of $\chi = 1.47$ rad, and the corresponding angle of refraction is $\beta = 0.714$ rad. "

How can k_{hi} be constant, when s varies ?

* The whole paragraph beginning with "Note that in our study, we have limited our analysis to the last nine fringes" added to answer Referee 2 remark has been written too fast and does not make sense

- in "we have chosen to define $H(s)$ based on fringes that we can directly observe" : on what else ?

- in "An alternative analogous definition for $H(s) = h(s, t)/[h_0 G(t)]$ could be constructed " : using your definitions $h_0 G(t) = h(0, t)$ so, this alternative definition seems to be the one actually used.

* In "Therefore if the weight of the tracer particle significantly influenced its velocity, it would be expected that the velocity of the silicone oil would be overestimated with the talcum powder tracers and

underestimated by the microbubble tracers."

The particle density is irrelevant for particle larger than the film thickness. For a particle of radius R , the comparison should be made between the mass of the displaced film = $(\rho_{\text{eau}} h \pi R^2)$ and the mass of the particle and of its associated meniscus.

Please provide the correct discussion

* In "It would be natural to select a boundary condition at the base of the bubble set by the rate at which liquid can be absorbed into the bath; however, it is questionable whether the assumptions of the thin-film equations would still be appropriate at this location. Instead, two boundary conditions can be obtained ..."

The word "instead" should be removed and replaced by "finally" : actually, I believe you need the bottom flux to close the problem, and the condition on H does not replace this missing condition. As the bottom flux is missing, one adjustable parameter consistently remains.

Reviewer #2 (Remarks to the Author):

First, I thank the authors for their careful reply, the introduction does more highlight the different cases associated to boundary conditions. Nevertheless, I am still wondering about the title, why do the authors suppress "viscous", I'm glad they precise bare but it seems like viscous is necessary as the Reynolds number is low. The universal drainage described in the paper might also sustained for larger Re but it has to be proven ... It would be of interest to calculate Re on page 4 like it is done for Bo . Thus I would like to recommend publication.

I also suggest three precisions that I write it in () :

- Last paragraph of page 1, 2nd sentence : still the (predicted) thickness profile (indeed thickness profiles were not measured in [2]).
- End of first paragraph in page 2 : (spatially) non-monotonic velocity.
- End of before last paragraph in page 5 : ... can be a hundred to a thousand times thicker near the base than at the top(, which it is not experimentally measurable with our interferometric technic but would be interesting to confirm in the future).

Responses to Reviewers:

We thank the Editor and Reviewers for their insightful feedback, which we have used to update our manuscript accordingly.

Below are point-by-point responses for each of the reviews.

Reviewer 1:

In the revised version, the connection to previous drainage models is improved and is now more explicit, but still needs clarifications.

The important contribution of the paper is to show that one former model, corresponding to the bare interfaces, was wrong in the literature, and to built the right one. It also provides comparison with other, valid, drainage models, with some interfacial stress at at least one interface.

In the present form, it is difficult for the reader to know which model of the literature is confirmed (the one with interfacial stress) and which one is proved wrong (the one without interfacial stress).

I will recommend the paper for publication if the following remarks are taken into account.

We thank the reviewer for their thoughtful feedback. Below are our responses to each remark.

1. *The introduction would be clearer if the wrong model of the literature was not presented as a fact. For example in*

"In these cases, the thickness-averaged drainage velocity u increases monotonically with inclination angle ϕ and depends on the boundary conditions at the two film surfaces."

Similarly in "Still, the thickness profile remains fairly uniform, with the midpoint and base of the bare bubble maintaining a relative thickness of 1.3 and 4 of that at the top [3]."

is it true ? false ? experimental ? theoretical ? for which case ?

My suggestion is that, at the various places where it is needed, the case with shear was discussed first, as a statement, and then the bare interface case with the experimental results (e. g. exponential decay at the apex) presented as robust, and the model (quasi uniform thickness, monotony of the velocity) presented using expressions as 'has been predicted/ found / assumed to be ...' or equivalent

We agree with the reviewer that a better distinction was needed between the robust results for shear flows and the more ambivalent results for elongational flows. We have followed up on the reviewer's suggested language and used phrases such as "predicted" and "assumed or found to be" to describe the theoretical results found in literature for elongational flows. Furthermore, we

strengthen the presentation of the robust results for the shear flows by using phrases such as “theoretical and experimental studies illustrate that...”.

2. *In "the film velocity will depend on the local thickness h and adopt a nearly uniform thickness profile" a word is missing*

We have modified the specific paragraph to highlight the different results on shear and elongational flows. As a result, the specific sentence was split into two.

3. *In "Importantly, all of these previous examples prescribed a drainage velocity that increased monotonically with polar angle."*

here again do not mix what is valid and what is not, what is measured and what is assumed.

We again thank the reviewer for pointing out our misconceived presentation of the established results of each flow. We have removed the particular sentence to avoid confusion. We elaborate on the different models in the same paragraph (see our response to point 4 below).

4. *"Because this drainage velocity is identical to the extensional flow proposed by Degregeas et al. [3], the universal bubble thinning expected for extensional drainage is recovered."*

This sentence is very unclear to me. For me the important messages are clearly identified in your rebuttal letter (answer 12) but not explicit here, especially the second point below.

In my opinion the discussion should be that :

- the limit " $b=R$ " leads to the Debregeas equations, with a reference to the sentence of the introduction "Here we show that the issue arises when tacitly assuming that velocity gradients across the film (shear) dominate velocity gradients along the film (elongation)." which is otherwise a little bit mysterious.

- unfortunately taking a limit of large b (or, much better, a small interface elasticity or viscosity) in the shear dominated equation, does not lead to the (proper) extension dominated equation (with a reference, that could be "The drainage of a foam lamella", Breward and Howell). So a very large b is not a physical limit.

As a personal note, I think the notion of slip length is nonsense for liquid-gas interfaces, simply because it should control a velocity difference between the two phases, and the gas phase velocity does not play any role in the dynamics, and is not even discussed. So I strongly support your fight against it.

We agree with the reviewer that the particular sentence is confusing and have modified it. After taking into account the two points above, we have added – in the same paragraph – the assumptions that lead to the analytical result for the velocity (i.e. when dissipation across the film dominates dissipation along the film). For the case of large slip (Debregeas), we highlight

that this assumption is not valid and provide the suggested reference as well as the “Drainage of a 2D bubble” by Howell, which includes the equations in the appendix.

5. *in "the film thickness continues to diverge at polar angles ..." I would say to increase*

We have changed “diverge” to “increase”.

6. *in "a result that is consistent with an increase in extensional viscosity due to strain hardening." If a strain hardening is expected or has been measured add a reference. Disjoining pressure is also a reasonable candidate.*

We agree that a reference was needed to support the notion of strain hardening and that the phrase “consistent with” was perhaps too strong. We have thus changed it to “could be attributed” and also added disjoining pressure as a possible mechanism with a corresponding reference.

7. *Fig 4a Inset: "In contrast to an extensional flow with two stress-free interfaces that cause a spatial divergence of the thickness, shear flows with one [15] or two no-slip interfaces [17, 18] lead to a fairly uniform thickness." please refer to the color used for each.*

We have changed the color of the curve and added the color references in the caption.

8. *Fig 3 f : the mention of the "incline flow" said in the rebuttal letter to have been removed is still there.*

We thank the reviewer for noticing this error. We have removed the label from the figure.

Supp Mat:

9. *" intensity is measured at an incident angle of $\chi = 1.47$ rad, and the corresponding angle of refraction is $\beta = 0.714$ rad. "*

How can k_{hi} be constant, when s varies ?

We appreciate that we could further clarify why the angle χ is constant in our measurements, and have added the following sentences to the supplemental description:

“In the projection illustrated in Fig. S1B, the incident angle χ changes with bubble arclength s . These points would correspond to a vertical line in the image taken by the camera. Meanwhile, the pixels along the edge of the captured image correspond to a great circle whose radial basis vector is orthogonal to the incoming light, or an orthodrome perpendicular to that shown in Fig. S1B. Thus for any arclength s on this great circle, the incident angle is $\chi = \pi/2$. We are unable to

measure the thickness directly when $\chi = \pi/2$ because the light does not pass through the film; instead, we measure the intensity a few pixels inside this great circle on a concentric circle with radius approximately 0.995 R (when projected on the image taken by the camera). For all points on this circle, the light strikes the bubble at an incident angle $\chi = \arccos(0.995) = 1.47$ radians.”

10. *The whole paragraph beginning with "Note that in our study, we have limited our analysis to the last nine fringes" added to answer Referee 2 remark has been written to fast and does not make sense*

- in "we have chosen to define $H(s)$ based on fringes that we can directly observe" : on what else ?

- in "An alternative analogous definition for $H(s) = h(s, t)/[h_0 G(t)]$ could be constructed " : using your definitions $h_0 G(t) = h(0, t)$ so, this alternative definition seems to be the one actually used.

We agree that this point was not particularly clear and rushed. We have changed the paragraph to elaborate how each function is computed and the corresponding effects of that choice on the range. The paragraph now reads as follows:

“Note that in our study, we have chosen to define $H(s)$ based on the fringes that we can directly observe. The maximum number of distinguishable fringes is nine, and thus limits the span of $H(s)$. Based on the ansatz $h(s, t) = h_0 H(s)G(t)$, we define $H(s) = h(s, t)/h(0, t)$, which we then use to compute $G(t) = h(s, t)/[h_0 H(s)]$. By including $H(s)$ in the calculations of $G(t)$, allows us to extend the possible range of $G(t)$ by approximately an order of magnitude larger than $H(s)$. An alternate approach would have consisted of first defining the temporal function as $G(t) = h(s, t)/h(s, 0)$ and then computing $H(s) = h(s, t)/[h_0 G(t)]$. This approach would have the benefit of extending the span of $H(s)$ at the expense of the span of $G(t)$. We have chosen to avoid this definition here, for one that is more traditional and direct.”

11. *In "Therefore if the weight of the tracer particle significantly influenced its velocity, it would be expected that the velocity of the silicone oil would be overestimated with the talcum powder tracers and underestimated by the microbubble tracers."*

The particle density is irrelevant for particle larger than the film thickness. For a particle of radius R , the comparison should be made between the mass of the displaced film = $(\rho_{\text{eau}} h \pi R^2)$ and the mass of the particle and of its associated meniscus.

Please provide the correct discussion

The referee raises an excellent point, which we now have included in our supplemental discussion. It is only at the higher angles – where the film is larger than the particle size – that we might expect these microbubbles to underestimate the drainage speed. At the lower angles, both particle types would be heavier than the displaced liquid and overestimate the drainage velocity. We thank the reviewer for highlighting this point and have made the requested change.

12. *In "It would be natural to select a boundary condition at the base of the bubble set by the rate at which liquid can be absorbed into the bath; however, it is questionable whether the assumptions of the thin-film equations would still be appropriate at this location. Instead, two boundary conditions can be obtained ..."*

The word "instead" should be removed and replaced by "finally" : actually, I believe you need the bottom flux to close the problem, and the condition on H does not replace this missing condition. As the bottom flux is missing, one adjustable parameter consistently remains.

We have modified the paragraph to make the point about the missing flux boundary condition more clear. In doing so, we have changed “instead” to “finally”.

Reviewer 2:

First, I thank the authors for their careful reply, the introduction does more highlight the different cases associated to boundary conditions. Nevertheless, I am still wondering about the title, why do the authors suppress “ viscous “, I’m glad they precise bare but it seems like viscous is necessary as the Reynolds number is low. The universal drainage described in the paper might also sustained for larger Re but it has to be proven ... It would be of interest to calculate Re on page 4 like it is done for Bo . Thus I would like to recommend publication.

We thank the reviewer for their recommendation and their additional feedback.

We have added the word viscous back into the title. Values of Re can be found in the legend of Fig. 3c.

I also suggest three precisions that I write it in () :

- 1. Last paragraph of page 1, 2nd sentence : still the (predicted) thickness profile (indeed thickness profiles were not measured in [2]).*
- 2. End of first paragraph in page 2 : (spatially) non-monotonic velocity.*
- 3. End of before last paragraph in page 5 : ... can be a hundred to a thousand times thicker near the base than at the top(, which it is not experimentally measurable with our interferometric technic but would be interesting to confirm in the future).*

We thank the reviewer for their suggestions. We have incorporated the bracketed phrases in our manuscript.

REVIEWERS' COMMENTS

Reviewer #1 (Remarks to the Author):

The authors have taken my comments into account.
I recommend the publication in the present form

Reviewer #2 (Remarks to the Author):

The authors has correctly include the remarks I asked, so I'm pleased to recommend publication of the paper "Universal non-monotonic drainage in large bare viscous bubbles"

Responses to Reviewers:

We thank the Reviewers for both recommending publication without any further questions or concerns.

Reviewer #1 (Remarks to the Author):

The authors have taken my comments into account.
I recommend the publication in the present form

Reviewer #2 (Remarks to the Author):

The authors has correctly include the remarks I asked, so I'm pleased to recommend publication of the paper "Universal non-monotonic drainage in large bare viscous bubbles"
Below are point-by-point responses for each of the reviews.